# Translating the Manufacture of Immunotherapeutic PLGA Nanoparticles from Lab to Industrial Scale: Process Transfer and In Vitro Testing

**DOI:** 10.3390/pharmaceutics14081690

**Published:** 2022-08-13

**Authors:** Maria Camilla Operti, Alexander Bernhardt, Jeanette Pots, Vladimir Sincari, Eliezer Jager, Silko Grimm, Andrea Engel, Anne Benedikt, Martin Hrubý, Ingrid Jolanda M. De Vries, Carl G. Figdor, Oya Tagit

**Affiliations:** 1Department of Tumor Immunology, Radboud Institute for Molecular Life Sciences, Radboud University Medical Center, 6500 HB Nijmegen, The Netherlands; 2Evonik Operations GmbH, Research Development & Innovation, 64293 Darmstadt, Germany; 3Institute of Macromolecular Chemistry CAS, Heyrovsky Square 2, 162 06 Prague, Czech Republic; 4Evonik Corporation, Birmingham Laboratories, Birmingham, AL 35211, USA

**Keywords:** drug delivery, PLGA, nanoparticles, nanomedicine, scale-up manufacturing, clinical translation

## Abstract

Poly(lactic-co-glycolic acid) (PLGA) nanoparticle-based drug delivery systems are known to offer a plethora of potential therapeutic benefits. However, challenges related to large-scale manufacturing, such as the difficulty of reproducing complex formulations and high manufacturing costs, hinder their clinical and commercial development. In this context, a reliable manufacturing technique suitable for the scale-up production of nanoformulations without altering efficacy and safety profiles is highly needed. In this paper, we develop an inline sonication process and adapt it to the industrial scale production of immunomodulating PLGA nanovaccines developed using a batch sonication method at the laboratory scale. The investigated formulations contain three distinct synthetic peptides derived from the carcinogenic antigen New York Esophageal Squamous Cell Carcinoma-1 (NY-ESO-1) together with an invariant natural killer T-cell (iNKT) activator, threitolceramide-6 (IMM60). Process parameters were optimized to obtain polymeric nanovaccine formulations with a mean diameter of 150 ± 50 nm and a polydispersity index <0.2. Formulation characteristics, including encapsulation efficiencies, release profiles and in vitro functional and toxicological profiles, are assessed and statistically compared for each formulation. Overall, scale-up formulations obtained by inline sonication method could replicate the colloidal and functional properties of the nanovaccines developed using batch sonication at the laboratory scale. Both types of formulations induced specific T-cell and iNKT cell responses in vitro without any toxicity, highlighting the suitability of the inline sonication method for the continuous scale-up of nanomedicine formulations in terms of efficacy and safety.

## 1. Introduction

At present, nanovaccines using nanoparticles (NPs) as vaccine vectors are becoming increasingly valuable tools to combat a plethora of diseases, such as cancer [1,2], hypertension [3], nicotine addiction [4], Alzheimer’s disease [5], bacterial [6,7,8] and viral infections such as the severe acute respiratory syndrome coronavirus (SARS-CoV-2) [9,10]. Polymeric NPs composed of biodegradable and biocompatible polymers such as poly(lactic-co-glycolic acid) (PLGA) have long shown promising potential for various biomedical applications, including the production of nanovaccines [2,7,8,11]. However, the clinical translation of PLGA-based nanoformulations can be an expensive and challenging process. Translational failures in method integration and technology transfer from lab- to large-scale manufacturing while complying with good manufacturing practice (GMP) regulations are considered among the main challenges [12]. Additionally, the physicochemical and structural properties of nanoformulations should be maintained during scale-up manufacturing as they directly determine the therapeutic efficacy and safety profiles [10,11,12,13]. Therefore, it is crucial to utilize a robust and scalable manufacturing technique from the earliest stages of development to ensure that the same high level of quality and reproducibility are achieved at large-scale production. To this end, continuous processes are considered the most favorable for nanomedicine production at a large scale because they offer the advantage of preferably terminating production at the desired scale without changing the process or formulation parameters [12,14].

We recently demonstrated the scale-up of an inline ultrasound-based process for the industrial production of PLGA-based nanoformulations that could reach a throughput of 84 g/h [15]. In the present study, we further adapt this technology to potentially scale-up the production of a clinical-stage nanovaccine formulation that was developed using a lab-scale batch sonication method and is currently tested in the Phase 1 clinical trial “Dose Escalation Study of Immunomodulatory Nanoparticles (PRECIOUS-01)” (ClinicalTrials.gov Identifier: NCT04751786) [1,2]. The nanovaccine formulation comprises three NY-ESO-1-derived antigen peptides (amino acid sequence 85–111, 117–143 and 157–165) and threitolceramide-6 (ThrCer6, also referred to as IMM60), a glycolipid α-galactosylceramide (α-GalCer) analog, encapsulated into PLGA nanoparticles. The PLGA NPs had a mean diameter of 150 ± 50 nm and a small polydispersity index (PDI < 0.2). These particle size characteristics were chosen according to their suitability for uptake by immune cells [16]. NY-ESO-1 is a testicular cancer antigen normally expressed in testicular germ cells and trophoblasts of the placenta [17] and in a wide range of cancers with a high incidence (around 25–30% of several cancers, such as melanoma (40%), lung (2–32%), bladder (32–35%) and ovarian (30%) cancer) [17]. IMM60 (ThrCer6) is a novel invariant natural killer T-cell (iNKT) agonist and dendritic cell transactivator, which enhances anti-tumor immune responses by inducing the secretion of a variety of pro-inflammatory cytokines, activating a broad spectrum of immune cells against the tumor [2,18,19,20,21].

For the scale-up manufacturing of the nanovaccine formulations with the specified characteristics, parameters of the inline manufacturing process were optimized. The particle size, PDI, zeta potential (ζ) as well as the drug loading and the release profile of each formulation produced using both batch and inline sonication modes are critically examined in a comparative manner. Finally, biological functional activity and dose toxicity of each particle type are evaluated in vitro. The reported process could translate the production of existing nanoformulations obtained with a batch sonication method at a lab scale to a large scale without modifying their efficacy and safety profiles.

## 2. Materials and Methods

### 2.1. Nanoparticle Production

RESOMER^®^ RG 502 H (PLGA) (lactide-to-glycolide mole ratio of 50:50, inherent viscosity 0.16–0.24 dL/g measured in Chloroform at 0.5 wt%) is an in-house product of Evonik Operations GmbH (Darmstadt, Germany). Dichloromethane (DCM) ≥ 99.5% was acquired from Avantor Performance Materials (Gliwice, Poland) and dimethyl sulfoxide (DMSO) 99.9%, USP grade, was procured from WAK-Chemie Medical GmbH (Steinbach, Germany). Poly(vinyl alcohol) (PVA) was purchased from Sigma-Aldrich (St. Louis, MI, US) and trehalose dihydrate was obtained from Carl Roth GmbH + Co. KG (Karlsruhe, Germany). NY-ESO-1-derived peptides, 85–111 (SRLLEFYLAMPFATPMEAELARRSLAQ), 117–143 (PVPGVLLKEFTVSGNILTIRLTAADHR) and 157–165 (SLLMWITQC), were custom-synthesized by GenScript Biotech (Piscataway Township, NJ, USA) with 97.3%, 72.1% and 96.7% purity levels, respectively. For the ease of reporting, we continue to refer to them as peptide 1 (85–111), peptide 2 (117–143) and peptide 3 (157–165). IMM60 was kindly provided by iOx Therapeutics Ltd. (London, UK).

#### 2.1.1. Laboratory-Scale Preparation of PLGA Nanoparticles Using Probe Sonication

##### Formulation Development

PLGA NPs with a targeted particle size of approx. 150 ± 50 nm and a PDI < 0.2 using the classic lab-scale probe sonication technique were prepared by dissolving the PLGA polymer in DCM achieving a concentration of 5 wt% (dispersed phase (DP)) and was subsequently emulsified together with an aqueous phase (continuous phase (CP)) containing 2 wt% PVA using a UP200St Ultrasonic Lab Homogenizer (Hielscher Ultrasonic GmbH, Teltow, Germany) equipped with a S26d2D needle probe. The treatment duration was set at 2 min. The process parameters were set at 100% amplitude and 100% phase. During the process, the sample container was kept immersed in an ice bath to prevent the degradation of sensitive material due to the high temperature generated by ultrasound. The obtained final suspension was diluted with MilliQ water in order to speed up the removal of the organic solvent and stirred for 1 h prior to particle size characterization (Table 1, exp. 1) [15].

##### Preparation of PLGA Nanovaccine Formulations Containing NY-ESO-1 Peptides and IMM60

In brief, 0.21 g of PLGA was dissolved into 3.99 g of DCM. The obtained solution was mixed with 0.42 mL of a 5 mg/mL solution based on the net peptide content of one of the three synthetic NY-ESO-1 peptides and 0.64 mL of a 0.5 mg/mL solution of IMM60 in DMSO. This DP containing the polymer, one of the NY-ESO-1 peptides, and IMM60 was emulsified together with 12.2 mL of the CP as abovementioned. Finally, the suspension was diluted with MilliQ water and stirred 1 h prior the downstream processes (Table 1, exp. 2). The identical protocol was also applied for the production of the placebo formulation with the exclusion of the addition of the Active Pharmaceutical Ingredients (APIs), i.e., NY-ESO-1 peptides and IMM60 (Table 1, exp. 3).

#### 2.1.2. Scale-Up Preparation of PLGA Nanoparticles Using Inline Sonication

##### Formulation Development

A GDmini2 Ultrasonic Inline Micro-Reactor (Hielscher Ultrasonic GmbH, Teltow, Germany) was used for the production of NPs in an inline mode [15]. In order to obtain comparable results, 15 mL of the same organic solutions at 5 wt% PLGA and 45 mL of the aqueous phase containing 2 wt% PVA were passed through the GDmini2 Ultrasonic Inline Micro-Reactor (Hielscher Ultrasonic GmbH, Teltow, Germany), line internal diameter 4 mm at a total flow rate (TFR) of 2 mL/min with a DP:CP flow rate ratio of 1:3. Prior to collection, another Tee junction was coupled to the sonicator outlet and MilliQ water (extraction phase [EP]) was pumped in through an ISCO pump (Teledyne ISCO, Lincoln, NE, USA). The pressurized coolant surrounding the glass cannula was maintained at a temperature of 10 °C to avoid damage of sensitive material during ultrasound treatment. Both the amplitude and phase of the indirect sonication process were kept at 100% (Table 2, exp. 1).

##### Scale-Up Preparation of PLGA Nanovaccine Formulations Containing NY-ESO-1 and IMM60

To compare the studies, the same DP described for the production of immunomodulating PLGA-based nanovaccines via probe sonication containing the polymer, one of the NY-ESO-1 peptides, and IMM60 was prepared and passed through the inline sonicator and emulsified together with the CP. Prior to collection, the EP was pumped in at a flow rate of 36 mL/min. Process parameters were kept unchanged as during the formulation development and are summarized in Table 2, exp. 2. In parallel, the same protocol without the addition of the APIs was applied for the manufacturing of the placebo formulation (Table 2, exp. 3).

### 2.2. Downstream Processes

The formulations were purified via tangential flow filtration (TFF) technique employing a KrosFlo^®^ KR2i TFF System (Repligen, Waltham, MA, USA). A Spectrum^®^ hollow fiber filter (D02-E750-05-N) of the MidiKros module family based of modified polyethersulfone (mPES) material was chosen with a molecular weight cut-off (MWCO) of 750 kD, fiber ID of 0.5 mm, 20 cm effective length (Repligen, Waltham, MA, USA). Samples were initially concentrated 6 times their volume and afterwards diafiltrated 5 times with MQ water. Subsequently, the purified suspensions were lyophilized in the presence of trehalose (3 vol% as final content) following the protocol described in our previous work [15]. Throughout the study, lyophilized particles were used for each test performed.

### 2.3. Analysis of Particle Size, PDI and Zeta Potential

Nanoparticulate systems produced in this work are designed to be administered parenterally. When starting with a lyophilized dosage form, water for injection (WFI) is generally used to re-suspend the formulations prior to administration to the patients. Therefore, the colloidal characterization of nanoparticles was performed in water. The mean particle size diameter as well as the PDI were determined via dynamic light scattering through a Zetasizer Nano ZS (Malvern Panalytical, Malvern, UK). As mentioned above, samples were previously diluted in sterile filtrated MilliQ water (0.2 µm) and measured three times at 25 °C with a 173° scattering angle. The surface charge of the NPs was investigated by ζ potential measurement at 25 °C using the same instrument using the Smoluchowski equation.

### 2.4. API Content Analysis

High-performance liquid chromatography (HPLC) composed of a DIONEX UltiMate 3000 Pump and Diode Array detector (UV-vis) (Thermo Fisher Scientific, Waltham, MA, USA) was utilized for the API content analysis.

Drug encapsulation efficiency (EE) and drug loading were verified following Equations (1) and (2):(1)$$\mathrm{Encapsulation}\;\mathrm{efficiency}\;\left(\%\right)=\frac{\mathrm{Weight}\;\mathrm{of}\;\mathrm{drug}\;\mathrm{found}\;\mathrm{in}\;\mathrm{the}\;\mathrm{nanoparticles}}{\mathrm{Weight}\;\mathrm{of}\;\mathrm{drug}\;\mathrm{initially}\;\mathrm{used}}\times 100$$
(2)$$\mathrm{Drug}\;\mathrm{loading}\;\left(\mathrm{mg}/g\right)=\frac{\mathrm{Weight}\;\mathrm{of}\;\mathrm{drug}\;\mathrm{found}\;\mathrm{in}\;\mathrm{the}\;\mathrm{nanoparticles}}{\mathrm{Weight}\;\mathrm{of}\;\mathrm{the}\;\mathrm{nanoparticles}}$$

#### 2.4.1. NY-ESO-1 Peptides

NY-ESO-1 peptide analysis was performed by HPLC using a Chromolith^®^ Performance RP-18e, 100 mm × 4.6 mm; 2 µm column (Waters Corp, Milford, MA, USA) with a mobile phase containing a gradient mixture of Solvents A and B. MilliQ water with 0.1 vol% trifluoroacetic acid (TFA) was used as Solvent A and acetonitrile with 0.1 vol% TFA was used as Solvent B. The gradient program (time/%B) was set as 0/5, 25/65, 35/95, 40/95, 45/5 and 50/5. The flow rate of the mobile phase was 1.0 mL/min. The column temperature was maintained at 45 °C and the chromatography was monitored at 220 nm. Injection volume was 10 µL. Retention time of the three NY-ESO-1 types was approx. 16 min for each peptide. Standard calibration solutions of NY-ESO-1 and samples were prepared in DMSO. The working concentrations were ≥20 µg/mL and the limit of detection (LOD) was approx. 5 µg/mL.

#### 2.4.2. IMM60

The IMM60 content of the NPs was determined by a Corona Veo Charged Aerosol Detector (CAD, Thermo Fisher Scientific, Waltham, MA, USA) coupled to the aforementioned HPLC system. The components of the formulation (PLGA, PVA, IMM60 and peptides) were separated by a XSelect CSH 18 column (130 Å, 2.5 µm, 3 mm × 150 mm) with VanGuard Cartridges (Waters Corp, Milford, MA, USA) coupled to a column heater (65 °C), eluents methanol–formic acid–triethylamine (99.0/0.05/0.05 vol%) with isocratic gradient flow rate at 1.0 mL/min followed by the detection of the components in the CAD system using an electrometer. The injection volume was 20 µL. The quantity of IMM60 was calculated by the interpolation of the standard calibration curves of IMM60 performed in the same way as for the NPs. Working concentrations were >40 µg/mL, while the LOD was determined as 3.7 µg/mL.

### 2.5. In Vitro Release of NY-ESO-1 Peptides

In vitro release studies were conducted over a 48 h period for NY-ESO-1 NPs formulations obtained at both lab scale and industrial scale using probe and inline sonication methods, respectively. Based on the drug loading estimated with HPLC measurements, particles containing 20 µg/mL of NY-ESO-1 of each formulation were suspended in 700 µL of 0.01 M PBS (pH 7.4) and processed using a Thermomixer (Eppendorf, Hamburg, Germany). The rotational speed was set to 500 rpm and the temperature to 37 °C. For each specified time interval, a separate sample was prepared, and the pellet as well as the total content (pellet plus supernatant) were analyzed to extrapolate the release behaviors. Prior to HPLC analysis, all the collected samples were lyophilized in order to remove the excess of water and subsequently solubilized in DMSO. Each drug release experiment was assessed in triplicate.

### 2.6. In Vitro Functional Biological Assays

#### 2.6.1. Antigen Presentation Assay

T cells transfected with mRNA of TCR recognizing NY-ESO-1 peptides were obtained following the protocol described by Dölen et al. [2]. Briefly, HLA-typed buffy coats obtained from the Sanquin blood bank (Nijmegen, Netherlands) were separated via Ficoll-Hypaque density gradient centrifugation method obtaining peripheral blood mononuclear cells (PBMCs). Monocytes and CD8+ T cells or CD4+ T cells were isolated, cryopreserved, and stored at cryogenic temperatures until use. Monocytes, once thawed, were cultured with interleukin-4 (IL-4; 300 IU/mL) and granulocyte macrophage-colony stimulating factor (GM-CSF; 450 IU/mL) to generate day 7 immature dendritic cells (DCs). On day 3, the cell culture medium was refreshed. Immature DCs were harvested on day 6, seeded in a 96 U-bottom plate (10,000 iDC/well) and cultured with NPs for 24 h. During this time, autologous T cells isolated from PBMCs were thawed and transfected with mRNA encoding the α and β chains of the TCR recognizing NY-ESO-1 peptide using electroporation. Subsequently, transfected T cells were added to the NPs–DCs culture mentioned above. Finally, to the CD8+ cells, lipopolysaccharides (LPS) were added. Supernatants were collected 72 h after establishment of the DC–T-cell co-cultures and analyzed for Interferon (IFN)-γ content by ELISA.

#### 2.6.2. iNKT Cell Activation

The ability of PLGA nanoformulations to activate iNKT cells was tested by studying the IL-2 production by the mouse iNKT cell hybridoma DN32.D3. The DN32.D3 cell line was cultured in RPMI 1640 medium with HEPES, supplemented with 1% glutamine, 10% Fetal Bovine Serum (FBS), 1% antibiotic–antimycotic (AA) solution and 50 mM β-mercaptoethanol. The mouse immature dendritic JAWS-II cell line was cultured in RPMI 1640 medium supplemented with 1% glutamine, 10% FBS, 1% AA solution and 5 ng/mL mouse GM-CSF. On day 0, the harvested JAWS-II cells were seeded in a 96 U-bottom plate (10,000 JAWS-II cells/well in RPMI 1640 medium supplemented with 1% glutamine, 10% FBS, 1% AA solution and 50 mM β-mercaptoethanol) and pre-cultured with different concentrations of IMM60 solution or nanoparticles containing IMM60 in equivalent concentrations at 37 °C and 5% CO_2_. After 24 h, 30,000 DN32.D3 cells/well were added to the JAWS-II DCs loaded with IMM60-containing particles or IMM60 solution. After that, supernatants were collected 48 h after the establishment of the co-cultures and analyzed for IL-2 secretion by ELISA.

##### ELISA

Two enzyme-linked immunosorbent assays were employed for the quantitative detection of human interferon gamma (IFN-γ) and mouse interleukin-2 (IL-2) produced during the incubation of the DC–T-cell co-cultures. Human IFN-γ uncoated ELISA and Mouse IL-2 Uncoated ELISA kits (Invitrogen, Waltham, MA, USA) were operated in accordance with the manufacturer’s protocol. Serum samples for IFN-γ and IL-2 analysis were diluted 1/5 and 1/30, respectively, in blocking buffer before adding them to the ELISA plates.

#### 2.6.3. Cytotoxicity Assay

A CellTiter-Glo^®^ (Promega Corporation, Madison, WI, USA) bioluminescence assay based on the measurement of adenosine 5′-triphosphate (ATP) with firefly luciferase was used to study cell viability after incubation with the different types of PLGA NPs. THP-1 cells (100 μL, 1 × 10^5^ cells/mL) were seeded into a 96-well flat-bottomed plate (white for luminescence measurement) and incubated for 24 h at 37 °C with 5% CO_2_ with 100 µL of a nanoparticle suspension containing the three NY-ESO-1 particle types together in equal content amounts, resulting in final NY-ESO-1 peptide sample concentrations ranging from 0.1 to 1 nmol/mL (µM) of total NY-ESO-1 peptide concentration. The maximum PLGA concentration reached per well was 0.7 mg/mL. Subsequently, CellTiter-Glo^®^ Reagent (Promega Corporation, Madison, WI, USA) was added to the wells (50 μL and 50 μL medium) and incubated at room temperature for 10 min protected from light. The luminescence was recorded using an Infinite^®^ M200 PRO (Tecan, Männedorf, Switzerland). Viability was determined in comparison to the cell culture media control set to 100% and presented as the percentage viability ± SD (standard deviation).

### 2.7. Statistical Analysis

Student’s *t*-test (two-tailed distribution, homoscedastic) (*n* = 3) was used during the particle characterization to determine the significance of the difference (*p* < 0.05) in size, PDI, ζ potential, encapsulation efficiency and drug loading, and in vitro tests among the compared groups with respect to the production scale (i.e., lab scale and industrial scale).

## 3. Results and Discussion

### 3.1. Assessment of Process and Formulation Parameters

Nano-sized PLGA formulations were developed at the lab scale using a batch technique based on probe sonication. This laboratory-scale technique is a popular approach due to the ease and adaptability of the operation, which allows for the rapid evaluation of formulations [12,15]. Adopting the process parameters summarized in Table 1, the obtained PLGA NPs (exp. 1) had a size of 188.0 ± 9.9 nm and the PDI was 0.11 ± 0.01, resulting in agreement with the required specification, i.e., size of 150 ± 50 nm and PDI < 0.2.

In need of a scalable method that exploits the same cavitation principle as the probe technique, an inline sonotrode was employed for the continuous production of such particles [15]. In this indirect method, the specimen flows nonstop inside of a glass cannula that is surrounded by a pressurized coolant that transmits the ultrasonic waves [15]. Since the sample is located inside disposable tubing all along the way, the entire process should be constructed aseptically, making it suitable for GMP-compliant processing. The particles generated following the process parameters summarized in Table 2 has a size of 175.7 ± 5.3 nm, while the PDI was recorded as 0.12 ± 0.01. Although the particle size was found to be slightly smaller compared to the particles obtained at the lab scale using probe sonication, these results were not significant (*p* > 0.05), suggesting that the technique was suitable for further study to scale-up nanomedicines for eventual commercialization.

### 3.2. Manufacturing and Characterization of PLGA Nanovaccine Formulations Containing NY-ESO-1 Peptides and IMM60

Three NY-ESO-1-derived peptides and the ThrCer6 (IMM60), a glycolipid α-galactosylceramide (α-GalCer) analog, were used as APIs to reproduce PLGA-based nanovaccine formulations that are currently tested in Phase 1 clinical trials “Dose Escalation Study of Immunomodulatory Nanoparticles (PRECIOUS-01)” [1,2]. The three epitope peptides were selected in the context of their respective human leukocyte antigen (HLA) alleles [2]. HLA corresponding to class I major histocompatibility complex (MHC) (A, B, and C) present antigens that attract CD8-positive [CD8+] T cells (also called as cytotoxic T cells), whereas those that correspond to the MHC class II (DP, DM, DO, DQ and DR) present to T-helper lymphocytes (also called CD4-positive [CD4+] or Th cells) [22]. When combined together, the three selected epitopes cover more than 80% of the European population for both class-I and class-II HLA alleles [2]. Table 3 lists the characteristics of each of the NY-ESO-1 peptides used in this study. Furthermore, IMM60 was included in the formulation as an adjuvant since it is a novel iNKT cell agonist and dendritic cell transactivator, which enhances T-cell responses [2,18].

For the encapsulation within PLGA nanoparticles, NY-ESO-1 and IMM60 were dissolved in DMSO and added to the chlorinated organic phase containing the polymer prior to sonication (Table 1, exp. 2; Table 2, exp. 2). Table 4 lists the characteristics of the placebo (NPs without any APIs) and NY-ESO-1 peptide/IMM60-loaded particles prepared on a large scale and at a lab scale.

Overall, particles produced on a large scale were smaller than the particles obtained by probe sonication at the lab scale, varying from 113 to 143 nm and 156 and 173 nm in size, respectively. The difference between both methods in terms of the obtained particle size was significantly different for the placebo, peptide 1 and peptide 2 formulations (Table 4). For both methods, particle size is determined by the size of emulsion droplets formed during sonication, which is directly dependent on the applied power per sample volume and the duration of sonication. Therefore, the measurement of the energy transmitted to the sample during sonication is crucial for the comparison of results. In inline sonication, a constant energy input of approx. 60 W was recorded when the device amplitude was set to 100%. In probe sonication, the energy recorded was about 36 W at the same amplitude. The total energy transmitted to the samples was evaluated as energy exerted during the operation time, which is the total duration of treatment (2 min) for the probe method, and the residence time in the glass tube (1.24 min) for the inline method. Thus, multiplying the recorded energy by the running time, a total energy of 4320 J and 4479 J were found to be transmitted to the samples in the probe and inline methods, respectively. In addition, the homogeneous exposure of the sample to cavitation forces enabled a more efficient mixing of phases in inline sonication, which can collectively account for the observed differences in particle size and size distribution. For both manufacturing methods, API encapsulation was observed to result in an increase in the particle size. Negative zeta potential (ζ) values that spanned a range from approx. −27 mV to −50 mV were observed for all particle types. The ζ difference of the various particles can be dictated by the isoelectric points (pI) of the peptides. PLGA placebo particles, containing free acidic functional groups having a pKa of about 5, have a recorded ζ of approx. −40 mV, confirming a negative charge in a neutral aqueous environment probably given by terminal carboxyls of the polyester chain. This value electrostatically assures the colloidal stability of the nanoparticle dispersion. The formulations containing peptide 1 (with a calculated pI of about 5.95) display a similar ζ around −35 mV. In contrast, the net positive charge of peptide 2 (pI approx. 9.18) partially shields the negative charges of PLGA and elevates the ζ to higher values, i.e., −28 mV. For the formulations containing peptide 3, which has the lowest pI of about 5.24, the most negative ζ was observed (approx. −48 mV).

The encapsulation efficiency (EE) of NY-ESO-1 peptides was tested via HPLC for each nanoparticle (NP). Peptide 1 and peptide 2 NPs had similar values located around 36% and 38%, respectively, while peptide 3 NP had an EE of approx. 20%. This diversity may be due to the difference in the grand average of hydropathicity (GRAVY) [23,24]. The GRAVY number of a protein is a measure of its hydrophobicity. The GRAVY value for a peptide or protein is calculated as the sum of hydropathy values [25] of all the amino acids, divided by the number of residues in the sequence [24]. When a protein is particularly hydrophobic or hydrophilic, it results in a relatively high or low hydropathy index. Usually, hydropathy values range from −2 to +2 for most proteins, with the positively rated proteins being more hydrophobic. Having a GRAVY of 0.115, 0.359 and 1.178 (calculated with ProtParam, ExPASy [23]) for peptide 1, 2 and 3, respectively, it can be noticed that the values are similar for the first two peptides, while there is a substantial difference compared to the third peptide. Since PLGA has a hydrophobic nature, it is likely to predict a higher EE for peptide 2 than for peptide 1, as indeed is revealed. Given the more hydrophobic nature of peptide 3, an even higher encapsulation efficiency compared to peptide 2 would be expected due to more favorable miscibility with the organic phase containing PLGA. However, when the length of peptides is considered (27 amino acids for peptide 1 and peptide 2), with only 9 amino acid residues, peptide 3 might be less firmly trapped in the PLGA mesh during particle hardening and escape during downstream processes. Indeed, also thermodynamically, a linear increase in the length of the interacting species (e.g., the length or number of hydrophobic peptide chains interacting with the number of polyester hydrogen bond donors/acceptors) leads to a linear increase in the molar interaction energy and an exponential increase in the interaction stability constant [26]. Therefore, a decrease in the EE of peptide 3 compared to the two longer peptides is to be expected, as has since been confirmed.

The EE of IMM60, which possesses a null formal charge, six hydrogen bond donors and seven hydrogen bond acceptors, was approx. 30% for all formulations regardless of the co-encapsulated peptide, indicating that both the production methods and the type of co-encapsulated peptide did not affect its EE.

Regarding the production volumes, the inline method was able to yield 1.6 g/h of API-loaded NPs, while the probe method could produce 0.21 g at a time, which is about 87% less than the inline sonication technique. On the other hand, the batch method is simpler from a handling point of view and can be repeated several times in a row. In addition, unlike inline processes, batch processes do not suffer from material loss due to dead volume, which must be considered especially in cases where the materials (e.g., APIs) are expensive. Within the inline apparatus described in this work, approx. 15 mL of dead volume was created [15]. Nevertheless, between batches, time spent on individual batch characterization activities, cleaning practices, as well as the cost of new equipment in the case of single-use materials must be taken into account, which are relevant drawbacks from a GMP manufacturing standpoint.

### 3.3. In Vitro Release Profiles of NY-ESO-1 Peptides

The in vitro release profiles of the NY-ESO-1 peptides were determined for all formulations using HPLC (Figure 1). For this study, particles were dispersed in PBS and the release profiles were monitored for 48 h. For each specific time point, a separate sample was prepared, and the pellet as well as the total content (pellet plus supernatant) were analyzed to extrapolate the release behaviors. The reasons for the evaluation of such a short time is dictated by the type of formulation. First, PLGA-based particles of ~150 nm administered intravenously are typically cleared from the bloodstream in about 48 h [16]. Second, once in the bloodstream, such immunological PLGA NPs are immediately taken up and processed by APCs (e.g., DCs) [1,2]. Therefore, a 48 h analysis window is suitable to assess the release behavior of such types of nanoparticles.

Each formulation type showed a distinct release profile that was strongly correlated with the encapsulated peptide. After an initial burst, the release overtime was zero for all particles. A clear difference in the release profile was observed for the formulations containing peptide 1 obtained at the lab scale and industrial scale (Figure 1A). An initial burst of ca. 60% of the total peptide content was observed for the lab-scale particles obtained by the probe sonication method, while it corresponded to approx. 75% for scale-up particles obtained via the inline method. This difference could be provoked by the different particle size as well as by the type of encapsulated peptide. In fact, as shown by Dutta et al. [27], larger PLGA-based NPs exhibit a lower burst release than identical but smaller particles. Additionally, it should be mentioned that peptide 1 possesses the lowest GRAVY, calculated as 0.115 (Table 3), which is indicative of peptides that tend toward hydrophilic behavior [25]. Therefore, it is likely that the smaller particles containing the same amount of peptides as the larger ones possibly have most of the peptide located close to or absorbed on their surface, which would explain the fast desorption and the tendency to release in aqueous environments. In line with this argument, the statistically significant difference in ζ potential between the two formulations (Table 4) could strengthen the idea that there is a different spatial arrangement of the peptide trapped in the particles.

The release profiles of peptide 2 and peptide 3 follow a similar trend for the formulations obtained at both scales (Figure 1B,C, respectively). The amount of peptide 2 released over time by the two types of particles nearly overlaps (Figure 1B). Both types of particles showed a release of about 30% that remained stable throughout the 48 h period. This could be due to the higher GRAVY value compared to peptide 1 that makes peptide 2 more alike to the hydrophobic PLGA polymer, increasing the binding intensity and thereby diminishing the burst effect. The peptide 3 release profile shown in Figure 1C can be identified as a middle ground between the two aforementioned peptide releases. Similar to peptide 1, an initial difference could be glimpsed between the two types of particles produced by the two different methods; however, after the first hour, the difference flattens out and the error bars overlap for all the time points analyzed (Figure 1C). As discussed earlier in the EE results, peptide 3 is the most hydrophobic one, which might suggest a better degree of entrapment. However, with only nine amino acids, the small size of peptide 3 can facilitate a fast release due to less stable interactions with the polymer compared to the other peptides.

It is also worth noting the release patterns: in all cases, the scale-up particles show a slight but gradual increase in peptide release in the first hours, while the initial release of the lab-scale particles is more stable. Overall, the differences in the release profiles of the particles produced by different methods should be taken into consideration as they could influence the results in biological systems. Therefore, functional studies were implemented to investigate how the physicochemical and release properties of the nanovaccine formulations influence their biological activity.

### 3.4. Antigen Presentation Assay

Antigen presentation is essential for adaptive immunity, where T cells recognize and kill pathogenic or pathogen-infected cells. Understanding the mechanisms of such immune responses is therefore important for the rational development and design of cancer vaccines. Antigen presentation is the expression of antigen molecules on the surface of APCs (e.g., DC), in association with MHC molecules [28,29,30]. Peptides presented by MHCs interact with the T-cell receptors, where MHC class II molecules present the antigen to a CD4+ helper T cell, whereas MHC class I molecules present it to CD8+ cytotoxic T cells. The prolonged interaction between a T-cell receptor and specific protein–MHC complexes eventually activates the T cells, which start producing cytokines such as interferon gamma (IFN-γ) [28,29,30]. In parallel, the recognition of the IMM60 by iNKT cells stimulates iNKT cells and APCs, which in turn induce the secretion of IFN-γ and various interleukins (ILs), activating a large pool of immune cells [20,21]. Therefore, in this study, the amount of IFN-γ and IL-2 produced are used as markers to assess the functional activity of the particles.

Figure 2 shows the dose/IFN-γ response curves obtained through co-culturing the DCs activated by various doses of PLGA nanovaccine formulations with the TCR-transfected T cells in vitro. Both types of nanoformulations produced at the lab and large scales are shown for all three peptides.

The response of CD8+ T cells against peptides 1, 2 and 3 are shown in Figure 2A–C, respectively. Regardless of the HLA types, the responses generated by the particles produced by the two methods were equivalent, indicating that nanoparticles retained their physicochemical properties upon scale-up.

The functionality of the iNKT cell agonist, IMM60, loaded into the PLGA nanoparticles was analyzed by comparing the two manufacturing methods in terms of inducing IL-2 production by the DN32.D3 hybridoma of mouse iNKT cells. Immature mouse JAWS-II DCs were loaded with the PLGA nanovaccine formulations and dose-dependent IL-2 production was assessed (Figure 3). In all cases, regardless of the production scale and manufacturing method used, the encapsulated IMM60 outperformed the solubilized free compound as also observed by Dölen et al. [20].

The formulations prepared at the lab scale (Figure 3A) performed similarly in terms of IL-2 production in comparison to the scale-up formulations (Figure 3B). The production of IL-2 appears to trace a sigmoid-like curve for the tested IMM60 concentrations ranging from 0.001 to 100 ng/mL. As the response to IMM60 is expected to be independent of the type of co-encapsulated peptide, the three curves derived from the peptide-containing particles shown in Figure 3A,B were pooled, and their averaged values were plotted for a better comparison of the lab-scale and scale-up formulations (Figure 4).

A paired *t*-test was exploited to inspect the difference in IL-2 production from mouse iNKT cells. Three of the six values evaluated were significant (*p* < 0.05); however, there was no clear predominance in the production of IL-2 generated by the particles produced by either method specifically. In fact, the lab-scale formulations were superior for two concentrations (0.1 and 1 ng/mL) and the scale-up formulations for the highest one (100 ng/mL). Additionally, even if significant differences were recorded for some concentrations, it must be highlighted that the general pattern of the dose–response curves is very similar and is in the same order of magnitude for each value, indicating that neither the manufacturing process nor the type of co-encapsulated peptide affect the functionality of the IMM60.

Overall, functional similarities have been identified throughout all the in vitro experiments, suggesting that nanovaccine formulations retain their activity upon scale-up.

### 3.5. Cytotoxicity Assay

A bioluminescence assay was used to measure the cell viability based on the quantitation of the ATP present with firefly luciferase, which is used as a marker for metabolically active cells. For this assay, THP-1 cells were selected. THP-1 is a human monocytic cell line derived from the peripheral blood of a childhood case of acute monocytic leukemia. These cells represent a valuable tool for investigating monocyte structure and function, and are often used as in vitro cancer cell models [31]. For this experiment, the three nanovaccine formulation types were mixed in equal content amounts, emulating the higher NY-ESO-01 peptide concentrations used in the antigen presentation assays, and tested to assess their possible in vitro toxicity in relation to their production scale. As a control, a mixture of the untreated APIs in the same concentration was tested. Since similar encapsulation efficiencies were obtained at both production scales for each peptide, a similar amount of PLGA particles was required to reach similar peptide contents. This PLGA NP content corresponded to approx. 0.7 mg/mL for each tested sample and was used as the concentration of the tested placebo NPs. The maximum total amount of sugar in the final suspension was approx. 0.5% (*w*/*v*), which was also tested in pure form as a further control. Figure 5 shows that scaling up the production of nanovaccine formulations did not affect the in vitro toxicity profile as the viability of THP-1 cells was not affected upon incubation with varying amounts of formulations prepared using different sonication methods.

## 4. Conclusions and Future Perspective

The main requirements for the clinical and commercial development of nanomedicines are high therapeutic efficacy and safety, along with the scalability of the manufacturing process. In this study, an inline sonication method was developed and adapted to scale up the production of PLGA-based nanovaccine formulations developed at the lab scale using a probe sonication method. Functional similarities were retained upon scaling up the production, emphasizing parallel formulation efficacy regardless of the production scale. Although further in vivo testing is necessary to provide a clear final statement, this study comprehensively demonstrated that the inline sonication method can reproduce the formulation and biological activity characteristics of PLGA-based nanovaccine formulations on a large scale for clinical and commercial development. Furthermore, despite the presence of dead volume that must be considered during the production phase, this continuous technique along with the previously established downstream processes can potentially be considered for GMP and aseptic manufacturing processes due to the application of fully enclosed tubes and containers that can be easily sterilized or replaced. For these reasons, the holistic manufacturing process developed in this study can be further exploited and adapted to the large-scale production of already-existing polymeric nanoformulations produced by the commonly operated lab-scale batch sonication technique. Taken as a whole, this work can be considered as a steppingstone for the industrial production of PLGA-based nanoformulations, paving the way for the manufacture and commercialization of future nanomedicines for the fight against cancer and beyond.

## Figures and Tables

**Figure 1 pharmaceutics-14-01690-f001:**
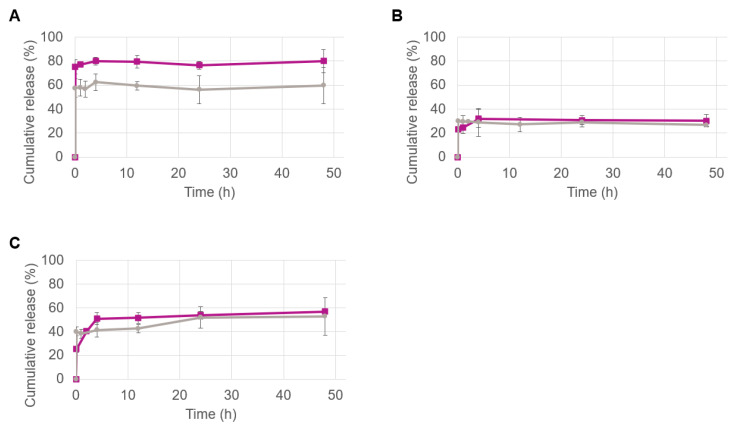
The cumulative release of the NY-ESO-1 peptide 1 (**A**), peptide 2 (**B**) and peptide 3 (**C**), from the PLGA nanoparticles obtained on an industrial scale (purple) and lab scale (grey) over time.

**Figure 2 pharmaceutics-14-01690-f002:**
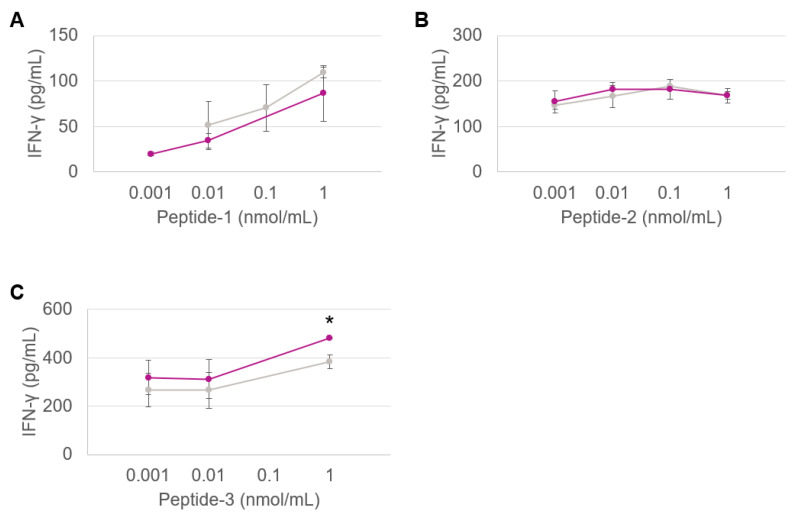
CD8+ T-cell response against peptide 1 presented by HLA-B7 (**A**), CD4+ T-cell response against peptide 2 presented by HLA-DRB1 (**B**) and CD8+ T-cell response against peptide 3 presented by HLA-A2 (**C**) contained in the PLGA nanovaccine formulations obtained via the inline (purple) and probe (grey) sonication methods. Each dot represents the mean value of triplicate wells. Assays are performed with three different nanoparticle batches. (*) Significant *p* < 0.05.

**Figure 3 pharmaceutics-14-01690-f003:**
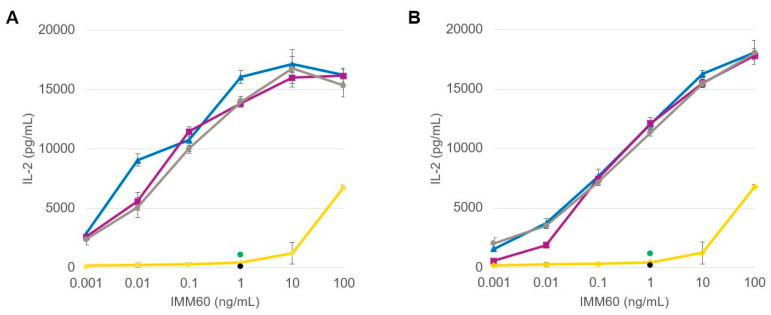
Dose-dependent IL-2 production by DN32.D3 mouse NKT cell hybridoma activated in vitro by PLGA nanovaccine formulations generated at the (**A**) lab scale and (**B**) industrial scale. Data obtained for particles containing peptide 1, peptide 2 and peptide 3 are represented in blue, purple and grey, respectively. Placebo particles, free soluble IMM60 and negative control are depicted in green, yellow and black, respectively.

**Figure 4 pharmaceutics-14-01690-f004:**
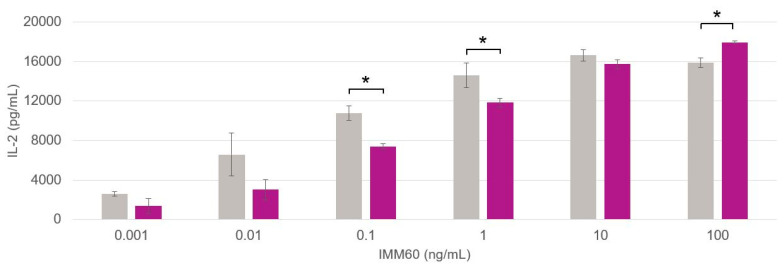
Differences in IL-2 production by mouse iNKT cells with respect to the particle production method. Average data points are acquired from the curves of peptide 1, peptide 2 and peptide 3 responses pooled together. Data obtained from particles generated at the lab -scale and industrial scale are depicted in grey and purple, respectively. (*) Significant *p* < 0.05.

**Figure 5 pharmaceutics-14-01690-f005:**
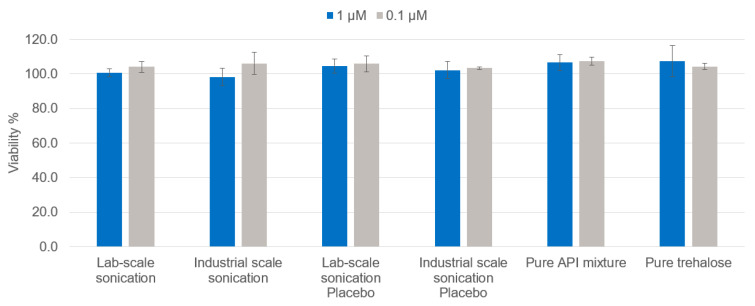
Viability of the THP-1 cells incubated with NY-ESO-1/IMM60 particles at different concentrations of the total peptide amount.

**Table 1 pharmaceutics-14-01690-t001:** Process parameters used for the formulation assessment and the production of immunomodulating nanovaccines with the probe sonication method.

Probe Sonication								
Exp. No.	API	DP Solution	DP Volume (mL)	CP Solution	CP Volume (mL)	Total Sonication Time (min)	EP Solution	EP Volume (mL)
**1**	Placebo	5 wt% PLGA, 95 wt% DCM	3	2 wt% PVA	9	2	MilliQ water	294
**2**	IMM60 and peptide 1 or peptide 2 or peptide 3	3.9 wt% PLGA, 74.3 wt% DCM, 0.006 wt% IMM60, 0.04 wt% NY-ESO-01, 21.7 wt% DMSO	4.06	2 wt% PVA	12.18	2	MilliQ water	290
**3**	Placebo	3.9 wt% PLGA, 74.4 wt% DCM, 21.7 wt% DMSO	4.06	2 wt% PVA	12.18	2	MilliQ water	290

**Table 2 pharmaceutics-14-01690-t002:** Process parameters used for the formulation assessment and the production of immunomodulating nanovaccines with the scale-up inline sonication method.

Inline Sonication								
Exp. No.	API	DP Solution	DP Flowrate (mL/min)	CP Solution	CP Flowrate (mL/min)	Residence Time (min)	EP Solution	EP Flowrate (mL/min)
**1**	Placebo	5 wt% PLGA, 95 wt% DCM	0.5	2 wt% PVA	1.5	1.24	MilliQ water	49
**2**	IMM60 and peptide 1 or peptide 2 or peptide 3	3.9 wt% PLGA, 74.3 wt% DCM, 0.006 wt% IMM60, 0.04 wt% NY-ESO-01, 21.7 wt% DMSO	0.5	2 wt% PVA	1.5	1.24	MilliQ water	36
**3**	Placebo	3.9 wt% PLGA, 74.4 wt% DCM, 21.7 wt% DMSO	0.5	2 wt% PVA	1.5	1.24	MilliQ water	36

**Table 3 pharmaceutics-14-01690-t003:** Characteristics of the NY-ESO-1 epitopes.

	Peptide 1	Peptide 2	Peptide 3
**Sequence**	SRLLEFYLAMPFATPMEAELARRSLAQ	PVPGVLLKEFTVSGNILTIRLTAADHR	SLLMWITQC
**Peptide position**	85–111	117–143	157–165
**No. amino acids**	27	27	9
**pI**	5.95	9.18	5.24
**GRAVY**	0.115	0.359	1.178
**HLA class**	Class I	Class II	Class I
**HLA type**	B7	DRB1	A2
**Epitope presentation to**	Cytotoxic T cells (CD8+)	T helper cells (CD4+)	Cytotoxic T cells (CD8+)

**Table 4 pharmaceutics-14-01690-t004:**
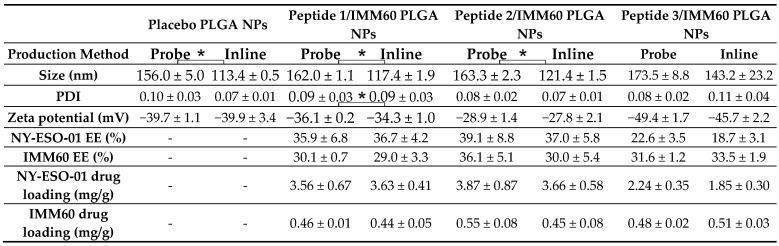
Nanoformulation characteristics obtained via direct probe batch and indirect inline continuous mode. Significance of the difference among the compared groups was determined with regard to the production method.

(*) Significant, *p* < 0.05.

## Data Availability

Not applicable.

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
