# Peer review of "Translating the Manufacture of Immunotherapeutic PLGA Nanoparticles from Lab to Industrial Scale: Process Transfer and In Vitro Testing"

_pharmaceutics, 2022, doi:10.3390/pharmaceutics14081690_

Round 1

Reviewer 1 Report

Title: Translating the manufacture of immunotherapeutic PLGA nanoparticles from lab- to industrial-scale: process transfer and in vitro testing

This is an experimental study based on the PLGA nanoparticles. The study has been done carefully and the experimental strategy seems meaningful. The experimental data is presented with statistical analysis. The scope of the study is vast and manuscript is also written well.

Some minor grammatical errors and spellings have been detected. The authors are requested to improve/correct all these faults before publication of this  article. 

The introduction part needs to be rewritten as it lacks coherency. It gives an impression of being taken from the dissertation or Ph.D. thesis. Lacks contextualization. The extra text should be removed to make it concise and relevant to the study.

Reviewer 2 Report

The manuscript submitted by Operti et al. deals with the preparation and characterization of PLGA nanoparticle (NP) with the scope of translating their manufacture from lab- to industrial-scale. The subject is relevant regarding the manufacturing industrial success of one the most studied nanoparticles.   

The manuscript is well structured and well written and presented methodologies have been disclosed previously by the research group and overall they fit on the scope of this manuscript. Before being accepted for publication, there are some some issues to be addressed by the authors, as follows:

-Taking into account that physicochemical characterization of nanoparticles was performed in water, how do authors explain any impact of culture medium and other relevant biological conditions on nanoparticle size and aggregation among other critical quality attributes;

-What is the rationale behind the use of rotational speed of 500 rpm in release studies? Did authors study other speed rates? In the same context, do authors consider a major influence of this experimental parameter?

-Release studies deserve a deep approach regarding the strong burst effect and incomplete release by APIs;

-It is not clear whether assays were performed with nanoparticles before or after lyophilization;

-Assuming that APIs quantification was not validated according to ICH guidelines authors are asked to provide concentration range, LOD among other parameters;

-In the in vitro cytotoxicity assay taking into account that authors tested placebo what is the rational of mentioning “The maximum PLGA concentration reached per well was 0.7 mg/mL? In this assay, authors are asked to increase rigor in methodology section as according to results there was no concentration range tested but two concentrations (0.1 and 1.0 nmol) tested instead. Besides there is lacking information regarding APIs and trehalose testing.  

-In line 334-335 it is stated “Based on these calculations, a total energy of 4320 J and 4479 J were found to be transmitted to the samples in the probe and inline methods, respectively.” How did authors make those calculations?

-In line 205 “The compounds were separated….” Can authors be more specific regarding those compounds;

- References need to be checked as at least 2 references, 17 and 19, are incomplete.
